# Reliability and Validity of Computerized Adventitious Respiratory Sounds in People with Bronchiectasis

**DOI:** 10.3390/jcm11247509

**Published:** 2022-12-19

**Authors:** Beatriz Herrero-Cortina, Marina Francín-Gallego, Juan Antonio Sáez-Pérez, Marta San Miguel-Pagola, Laura Anoro-Abenoza, Cristina Gómez-González, Jesica Montero-Marco, Marta Charlo-Bernardos, Elena Altarribas-Bolsa, Alfonso Pérez-Trullén, Cristina Jácome

**Affiliations:** 1Health Sciences Faculty, Universidad San Jorge, 50830 Zaragoza, Spain; 2Hospital Clínico Universitario Lozano Blesa, 50830 Zaragoza, Spain; 3Instituto de Investigación Sanitaria (IIS) Aragón, 50830 Zaragoza, Spain; 4Research Group Movimiento Humano, Universidad de Zaragoza, 50830 Zaragoza, Spain; 5Hospital General de la Defensa, 50830 Zaragoza, Spain; 6Center for Health Technology and Services Research (CINTESIS), Faculty of Medicine, University of Porto, 4099-002 Porto, Portugal

**Keywords:** bronchiectasis, adventitious respiratory sounds, crackles, wheezes, reliability, validity

## Abstract

Background: Computerized adventitious respiratory sounds (ARS), such as crackles and wheezes, have been poorly explored in bronchiectasis, especially their measurement properties. This study aimed to test the reliability and validity of ARS in bronchiectasis. Methods: Respiratory sounds were recorded twice at 4 chest locations on 2 assessment sessions (7 days apart) in people with bronchiectasis and daily sputum expectoration. The total number of crackles, number of wheezes and wheeze occupation rate (%) were the parameters extracted. Results: 28 participants (9 men; 62 ± 12 y) were included. Total number of crackles and wheezes showed moderate within-day (ICC 0.87, 95% CI 0.74–0.94; ICC 0.86, 95% CI 0.71–0.93) and between-day reliability (ICC 0.70, 95% CI 0.43–0.86; ICC 0.78, 95% CI 0.56–0.90) considering all chest locations and both respiratory phases; wheeze occupation rate showed moderate within-day reliability (ICC 0.86, 95% CI 0.71–0.93), but poor between-day reliability (ICC 0.71, 95% CI 0.33–0.87). Bland–Altman plots revealed no systematic bias, but wide limits of agreement, particularly in the between-days analysis. All ARS parameters correlated moderately with the amount of daily sputum expectoration (r > 0.4; *p* < 0.05). No other significant correlations were observed. Conclusion: ARS presented moderate reliability and were correlated with the daily sputum expectoration in bronchiectasis. The use of sequential measurements may be an option to achieve greater accuracy when ARS are used to monitor or assess the effects of physiotherapy interventions in this population.

## 1. Introduction

Bronchiectasis is a chronic respiratory syndrome associated with a recurrent airway inflammation-infection vicious cycle that originates bronchial dilatations and chronic symptoms [1]. In a survey, people with bronchiectasis rated sputum production as the most severe symptom, followed by fatigue, breathlessness and cough [2]. These symptoms, together with the frequent acute exacerbations, negatively impact on the quality of life of people with bronchiectasis [3] and create a significant health burden. A systematic review conducted by Goeminne P.C. et al. found that mean annual age-adjusted hospitalization rates in bronchiectasis range from 1.8 to 25.7 per 100,000 population, with an average hospitalization duration between 2 and 17 days [4]. Moreover, the estimated annual management cost is around EUR 3515–EUR 4672 in Spain, being higher in those with >2 exacerbations/year and *Pseudomona aeruginosa* infection [4]. Therefore, close monitoring of respiratory status in people with bronchiectasis is recommended [5,6].

Computerized lung auscultation is a simple, objective and non-invasive technique used by healthcare professionals to monitor the respiratory system and evaluate changes after interventions [7,8]. Adventitious respiratory sounds (ARS), such as crackles and wheezes, are associated with important airway mucus accumulation and bronchial obstruction [7]. The number and distribution of crackles are related to physiological measures of disease severity in idiopathic pulmonary fibrosis [9]. Inspiratory crackles heard over the basal parts of the lungs are related to chronic obstructive pulmonary disease (COPD) [10,11]. Both crackles and wheezes have been proposed as coadjutant measures to diagnose and monitor exacerbations [12,13] and a sensitive outcome to detect changes after a pulmonary rehabilitation program in COPD [14]. In people with bronchiectasis, ARS are feasible outcomes for assessing the effects of a crucial component of pulmonary rehabilitation in this population, the airway clearance techniques [15], but their measurement properties are not fully known.

The reliability and validity of computerized ARS have been studied in people with COPD [16,17]. Overall, the acceptable within-day and between-day reliability of crackles (mean number) [16,17] and wheezes (mean number [16], occupation rate [17]) were reported. In this population, computerized ARS had negligible-to-moderate correlations with lung function and low positive correlations with patient-reported outcome measures [17]. Yet, in people with bronchiectasis, computerized ARS have been poorly explored. Marques A. et al. [18] showed good-to-excellent within-day reliability for crackles (the crackle initial deflection width and two-cycle deflection) in a mixed population of people with cystic fibrosis and bronchiectasis. Nevertheless, between-day reliability and validity of ARS in people with bronchiectasis were not explored. The knowledge of these psychometric properties is crucial before the widespread use of computerized ARS in real-world clinical practice.

Therefore, the aim of this study was to determine the within-day and between-day reliability and validity of computerized ARS (crackles and wheezes) in people with stable bronchiectasis.

## 2. Materials and Methods

### 2.1. Study Design

A cross-sectional study was conducted with outpatients with stable bronchiectasis recruited from two tertiary hospitals. The study was conducted at San Jorge University (Zaragoza, Spain) between October 2016 and December 2018. Reliability and validity were analysed according to COnsensus-based Standards for the selection of health Measurement INstrument (COSMIN) taxonomy [19] and Guidelines for Reporting Reliability and Agreement Studies (GRRAS) [20]. The study protocol was approved by the Aragon Ethics Committee (PI14/00100), and all participants signed the informed consent before any clinical data were recorded.

### 2.2. Participants

Patients were invited to participate if they were aged 18 years or over, had a primary diagnosis of bronchiectasis using a high-resolution computed tomography scan [21,22], had chronic sputum expectoration and were clinically stable (no hospital admissions, exacerbations or changes in respiratory system medication) [23] in the last four weeks. Patients were excluded if they had a diagnosis of cystic fibrosis; were smokers or ex-smokers (≥10 pack/year); presented significant chest wall deformities, a history of lobectomy/pneumonectomy or neuromuscular disorders; or needed continuous treatment with non-invasive ventilation. Concomitant respiratory conditions were registered from medical records. The pulmonologists identified possible participants, explained the study purpose and asked about their willingness to participate. When participants agreed, the pulmonologists gave them two containers to collect daily spontaneous sputum expectoration in the two days before the study visit. Then, researchers contacted them by phone and explained the study and respective procedures in more detail and reminded them to collect their daily sputum.

### 2.3. Data Collection

Participants were asked to attend two assessment sessions over two consecutive weeks (one session per week). The two sessions were scheduled for the same weekday and time. Medical records were reviewed to extract information about the aetiology of the bronchiectasis, radiological severity, chronic airway infection, exacerbations in the last year and previous hospital admissions. Sociodemographic data (age, gender) and chronic respiratory medication were reported by the participants in the first session. Lung function (forced spirometry) [24], sputum volume collected in the containers (average of 24 h sputum volume obtained in the two previous days) [25], the impact of coughing (Leicester Cough Questionnaire, LCQ) [26], quality of life (quality of life-bronchiectasis, QoL-B) [27] and dyspnea (MRC Breathlessness Score) [28] were also assessed at the beginning of the first session. Disease severity was calculated using the Bronchiectasis Severity Index (BSI) [29].

Respiratory sounds were recorded at both sessions following the same procedure. During the recording period, participants were sitting and breathing through a mouthpiece connected to a pneumotachograph (Spiro USB spirometer, CareFusion, Kent, Basingstoke, UK) that allowed gaming visual feedback of airflow during both inspiratory and expiratory phases (Figure 1). During recordings, participants were instructed to maintain an airflow of 0.4–0.6 L/s (similar to spontaneous breathing), as this airflow range was found to be reliable in other respiratory conditions [16].

Two trained physiotherapists recorded the respiratory sounds at four standardized chest locations (i.e., right and left: anterior and posterior chest) [30] (Figure 1) with two electronic stethoscopes (Littmann^®^ 3200, 3M, Saint Paul, MN, USA). Three additional locations (trachea, right and left lateral chest) are proposed by CORSA (Computerized respiratory sound analysis) [30] guidelines, but researchers decided not to record these locations based on their clinical judgment on the usefulness of these locations in patients with bronchiectasis. In fact, the trachea is rarely used in clinical practice and has shown poor reliability [16], and lateral chest recordings usually present low quality due to excessive artifacts. Recordings were made for 25 s simultaneously at the right and left chest locations, first in the posterior and then in the anterior chest. The whole process was repeated after a 1 min interval. Participants were advised to avoid coughing during or between the recording periods. During recordings, the respiratory sounds were transmitted directly to a computer via Bluetooth^®^ and stored in .wav format.

The same procedure was repeated in the second session. Therefore, each location was recorded four times in total (two times in each session) (Figure 1).

### 2.4. Respiratory Sounds Analysis

Respiratory phases (inspiration and expiration) were detected manually by one trained researcher (MF-G) and then revised by another trained researcher (BH-C). Computerized ARS (crackles and wheezes) were identified and analysed automatically through validated algorithms [31,32] implemented in MATLAB R2013b (MathWorks, Natick, MA, USA). The parameters extracted from the sound files were mean number of crackles, mean number of wheezes and wheeze occupation rate (%). Each parameter was extracted per respiratory phase (inspiration and expiration) and complete respiratory cycle.

### 2.5. Statistical Analysis

Considering a minimum intraclass correlation coefficient (ICC) of 0.9 with a 95% confidence interval (95%CI) of 0.2 (α = 0.05, κ = 2), a minimum of 25 participants were required to assess the reliability [33] of ARS, given an anticipated dropout rate of 20%.

Variables were checked for normality using the Shapiro–Wilk test. Data were described using absolute and relative frequencies for categorical variables, mean (standard deviation, SD) for continuous variables normally distributed or median [percentile 25–percentile 75, P_25_–P_75_] for those not normally distributed. The characteristics of the computerized ARS (crackles and wheezes) were described for each of the two recordings performed in each assessment session, stratified for anterior and posterior chest location, and pooled for right and left sides. 

Within-day reliability of computerized ARS was estimated from the two recordings collected in the first assessment session, while between-day reliability was analysed from the average of the values obtained in the two assessment sessions. Within-day and between-day reliability was calculated using the ICC_3,k_ (a two-way mixed effects, absolute agreement, multiple measurements) [34] with a 95% CI and interpreted as excellent (>0.75), moderate (0.4–0.75) or poor (<0.40) based on the value of the lower limit of the CI [35]. The level of agreement for within-day and between-day measures was represented and explored using the Bland–Altman method, including their 95% CI for systematic bias [36]. The standard error measurement (SEM) and the smallest detectable change (SDC) were also calculated using the following equations [37].: SEM = SD baseline × √(1 − ICC)
and
SDC = 1.96 × √(2 × SEM) 

The mean values obtained from recordings performed in both assessment sessions were used to evaluate the construct (convergent) validity of ARS. The relationship between ARS and lung function (percentage of the predicted forced expiratory volume in one second, FEV_1_% pred.), 24 h sputum volume, cough severity (LCQ), quality of life (QoL-B), radiological severity (number of lobes affected) and BSI was assessed using Spearman’s rank correlation coefficient. The strength of the correlations was interpreted as weak (rho ≤ 0.29), moderate (0.30 < rho > 0.59) or strong (rho ≥ 0.60) [38].

A *p* value < 0.05 was considered statistically significant for all analyses. Data were analysed using SPSS v.24.0 (IBM, Chicago, IL, USA), and the plots were created using GraphPad Prism 5.01 (GraphPad Software, La Jolla, CA, USA).

## 3. Results

A total of 28 participants were included in the within-day reliability analysis. The characteristics of the participants are described in Table 1. Most participants were female (*n* = 19; 68%) with a mean age of 62 (SD 12) years and a median BSI of 7.5 [P_25_–P_75_ 5.0–12.0]. Only 25 participants were included in the between-day reliability and in the convergent validity analysis because two participants did not attend the second assessment session (no reason provided) and one participant had an acute exacerbation.

### 3.1. Crackles

A summary of crackles’ characteristics obtained during the two assessment sessions is presented in Appendix A.

#### 3.1.1. Within-Day Reliability

Overall, within-day reliability for all chest locations was moderate for the total number of crackles during expiration and the complete respiratory cycle and poor during inspiration (Table 2). At the anterior chest, moderate within-day reliability was shown in the total number of crackles during inspiration and the complete respiratory cycle; however, poor within-day reliability was observed during expiration. At the posterior chest, within-day reliability for the total number of crackles was moderate during the complete respiratory cycle, excellent during expiration and poor during inspiration. No significant bias was observed in the Bland-Altman plot, and the limits of agreement seem to be narrowed when the average of the number of crackles is less than seven (Figure 2). Considering all chest locations, the range found for SEM was 0.58–1.26 and SDC 2.12–3.11, with inspiratory crackles showing the lowest values for both parameters and total crackles during the complete respiratory cycle showing the highest values.

#### 3.1.2. Between-Day Reliability

Considering all chest locations, between-day reliability for the total number of crackles was moderate during inspiration and the complete respiratory cycle and poor during expiration (Table 2). Between-day reliability was poor for the total number of crackles at the anterior chest during expiration and the complete cycle, but moderate during inspiration. At the posterior chest (during inspiration and the complete cycle) a moderate between-day reliability was observed. No bias was observed for crackles using the Bland–Altman plot, but the limits of agreement were slightly wide, especially if the average of the number of crackles exceeded seven (Figure 2). Considering all chest locations, the range found for SEM was 0.46–1.81 and SDC 1.89–3.73, with total crackles during both respiratory phases showing the highest values and inspiratory crackles showing the lowest values.

#### 3.1.3. Construct Validity

Moderate positive correlations were observed between the total number of crackles considering the complete respiratory cycle and 24 h sputum volume. The total number of inspiratory crackles also correlated moderately and positively with 24 h sputum volume. No significant correlations were found between crackles and FEV_1_, LCQ, QoL-B, number of lobes affected and BSI (Table 3).

### 3.2. Wheezes

Appendix A summarises the wheezes’ characteristics extracted during the two assessment sessions, and the estimated reliability parameters (ICC, SEM, SDC) are presented in Table 3.

#### 3.2.1. Within-Day Reliability

Globally, moderate within-day reliability was observed for the number and occupation rate of wheezes during inspiration, expiration and the complete respiratory cycle when considering all chest locations and each chest region separately (Table 3). The only exception was the poor reliability of the inspiratory occupation rate at the posterior chest. The Bland–Altman plot showed no systematic bias for the number of wheezes and the occupation rate, but with wide limits of agreement (Figure 2). Considering all chest locations, SEM ranged from 0.61 to 8.88 and SDC from 2.17 to 8.26, with inspiratory wheezes showing the lowest values for both parameters and the occupation rate during expiration showing the highest values.

#### 3.2.2. Between-Day Reliability

Between-day reliability of the number of wheezes at both chest regions and at the anterior chest was moderate during expiration and the complete respiratory cycle, but poor during inspiration (Table 3). Globally, poor reliability was observed for the number of wheezes at the posterior chest. Overall, the occupation rate showed poor between-day reliability, except for the expiratory phase at the posterior chest region. The Bland–Altman plot revealed no systematic bias, but wide limits of agreement (Figure 2). The range found for SEM was 1.02–11.52 and SDC 2.81–9.41, considering all chest locations, with inspiratory wheezes showing the lowest values for both parameters and the occupation rate during expiration showing the highest values.

#### 3.2.3. Construct Validity

The number of wheezes and the occupation rate correlated moderately and positively with the 24 h sputum volume during inspiration, expiration and the complete cycle. No other significant correlations were observed (Table 4).

## 4. Discussion

In people with stable bronchiectasis, the total number of crackles and wheezes showed moderate within-day and between-day reliability, considering all chest locations and both respiratory phases; however, the occupation rate only showed moderate reliability in the within-day analysis. In addition, the total number of crackles, number of wheezes and the occupation rate in all chest locations and during the complete respiratory cycle correlated moderately with the amount of 24 h sputum expectoration in this population.

The ARS reliability values observed in this study are in line with data previously reported in other respiratory diseases. Globally, the most reliable parameters tend to be achieved when all chest locations and both respiratory phases are considered. Likewise, the within-day reliability of ARS tends to show higher values than the between-day reliability.

To the best of our knowledge, the only previous study analysing the reliability of computerized ARS in bronchiectasis was conducted by Marques et al. [18]. This study found moderate to excellent within-day reliability for two crackles’ parameters, the initial deflection width and the two-cycle deflection, in a mixed sample of people with bronchiectasis and cystic fibrosis. Unfortunately, no specific data were available for the bronchiectasis subgroup, despite the need to identify bronchiectasis as a distinct disease from cystic fibrosis [39]. This was one of the main reasons for excluding people with cystic fibrosis in the present study. Moreover, the crackles parameters reported in this study are rarely used in clinical practice and in trials evaluating the effects of interventions, as humans cannot recognise these parameters being totally dependent on the available algorithms. In fact, the number of crackles is a parameter more commonly reported in previous studies evaluating psychometric properties of computerized ARS [16,17]. For these reasons, we believe that the total number of crackles is a more useful parameter to evaluate treatment effects and monitor patients in the long run (e.g., helping in the diagnosis of interstitial lung disease [40], monitoring exacerbations in people with COPD [13], assessing the effects of pulmonary rehabilitation [14]). Specifically in bronchiectasis, the number of crackles demonstrated to be feasible for assessing the effects of airway clearance techniques and was suggested as a good primary endpoint in trials evaluating components of pulmonary rehabilitation in this population.

A previous study conducted in people with COPD [17] showed that the number of crackles appear to be a more reliable parameter during inspiration, particularly at the posterior chest, than during expiration in the within-day and between-day analysis. Our data suggest that this pattern may be similar in people with bronchiectasis, as between-day reliability, taking in consideration all chest locations, was greater for inspiratory crackles than expiratory crackles. In fact, the greater consistency of inspiratory crackles over time found in this study may be related to the fact that this ARS parameter is mainly dependent on the pathophysiology of the surrounding tissue, and both chronic respiratory diseases, COPD and bronchiectasis, are characterised by thickening of the bronchial wall and loss of bronchial support, leading to intermittent airway opening-closing during normal breathing [7,41]. However, unlike in people with COPD [17], our data did not suggest better reliability in a specific chest location. This may be explained by the heterogeneity in the extent of the disease and clinical presentation in people with bronchiectasis [42,43].

Wheezes are associated with airflow limitation in abnormally narrowed or compressed airways [44]. The presence of inspiratory wheezing is commonly associated with more severe airway obstruction [44], and expiratory wheezing is more related to the dynamic collapse of airways (e.g., after forced expiratory maneuvers) [45] and even the presence of secretions [7]. Data from Oliveira et al. [17,46] suggested that wheezes are not an adequate ARS parameter in people with COPD due to their strong dependence on airflow and high variability, especially observed during expiration. However, our findings demonstrate that wheezes during expiration and the complete respiratory cycle had moderate reliability in our population, considering both chest locations, and the reliability values obtained for wheezes and the occupation rate at the anterior and posterior chest locations were slightly better during expiration than inspiration. This difference may be related to the fact that (i) in our study, the airflow was monitored during ARS recordings and forced expiratory maneuvers were not allowed between the recordings to reduce their impact on the findings; (ii) the clinical impact of the inspiratory wheezes may be less relevant in our study because our sample presented less severity of bronchial obstruction (FEV_1_ = 49% in the Oliveira et al. [17] study vs. FEV_1_ = 77% in our study); (iii) expiratory wheezes were clearly more predominant than inspiratory wheezes in our sample, maybe associated with the fact that all of our participants presented daily sputum expectoration as a common clinical feature of the bronchiectasis disease.

No systematic differences have been found for the number of crackles and wheezes in the within-day and between-day analysis as it was expected in a stable condition; however, the mean difference between the measurements was large for the occupation rate in both analyses. The limits of agreement were quite wide for all parameters, particularly in the between-day analysis; the only exception was when the average of the crackles did not exceed seven. Therefore, the use of repeat measurements may be a good solution to improve the dispersion around the mean, as previously was reported for other outcome measures [25,47]. Overall, SEM and SDC values tend to be high for all ARS parameters, particularly in the between-days analysis, as it was expected. It is known that both parameters depend on the population studied [48] and, thus, comparisons are difficult. SEM and SDC are essential values to help in the interpretation of the tools; therefore, in clinical practice or in future studies using ARS as an outcome measure in bronchiectasis, the data provided in this study can be used to support the interpretation of findings.

Daily spontaneous sputum expectoration is the only clinical outcome that correlates moderately with the number of crackles, wheezes and occupation rate in our study, in contrast with a previous study conducted on COPD that found significant correlations between the number of crackles at the posterior chest location and lung function (FEV_1_%). As stated above, the less severity of bronchial obstruction in our study may explain why a correlation was not found. Future studies with patients with greater decline of lung function will clarify this issue. Chronic sputum production is a common clinical feature of people with bronchiectasis, and it is known that crackles and wheezes may be related to the presence of excess of mucus in the airways [7]. In fact, a positive moderate correlation was observed between the number of crackles during the expiratory phase and the ratio of sputum expectorated during an airway clearance session in bronchiectasis [15]. Our data suggest that ARS may not be related to disease severity (BSI, FEV_1_%, number of lobes affected) and patient-reported outcomes (LCQ and QoL-B) and, therefore, the use of ARS to monitor people with bronchiectasis or assess the effects of treatments (e.g., airway clearance techniques, pulmonary rehabilitation) should be encouraged in combination with other measurements. Future studies evaluating if other respiratory sounds parameters, such as intensity or frequency, present a better relationship with clinical outcome measures, and if ARS change along the disease trajectory (e.g., early vs. last stage) and across different clinical statuses (e.g., exacerbation vs. stability) using matched controls in bronchiectasis are recommended.

### Limitations

Although we have estimated the sample size needed to determine the reliability of ARS, this estimation was below the threshold of 50 participants recommended by the COSMIN initiative [19]. Future studies should estimate the sample needed and follow COSMIN to increase the robustness of the results. This sample size also limited the possibility to perform a sensitive analysis to identify if participants with greater radiological severity tend to present similar findings. The eligibility criteria, mainly the exclusion of smokers, also limited the generalisation of the results to this specific population. CORSA guidelines recommend recording respiratory sounds at seven standardized locations [30]. In our study, we used four of these standardized locations, which facilitated comparison with findings from other conditions [16,17]. Recordings were analysed using automatic validated algorithms with acceptable accuracy; nevertheless, automatic annotations are not free from errors (artifacts can be wrongly detected as ARS, and other times, ARS may be missed) [31]. To improve the detection process, two trained researchers independently reviewed all automatic detections to make necessary changes. Nevertheless, human annotation is also susceptible to error. More robust algorithms to detect ARS need to be developed. Finally, our sample size limits the possibility of performing a sensitive analysis to identify if participants with greater radiological severity tend to present similar findings.

## 5. Conclusions

Our data suggest that the number of crackles and wheezes present moderate reliability at short time intervals and are moderately correlated with the daily amount of sputum expectorated in people with stable bronchiectasis. However, the level of agreement, the SEM and SDC provide information on the presence of high variability for all ARS parameters. Therefore, the use of sequential measurements when ARS are evaluated, including all chest locations and both respiratory phases, is recommended to achieve greater accuracy and complete the clinical assessment with other outcome measures when ARS are used to monitor or assess the effects of interventions, such as airway clearance techniques or pulmonary rehabilitation, in people with stable bronchiectasis.

## Figures and Tables

**Figure 1 jcm-11-07509-f001:**
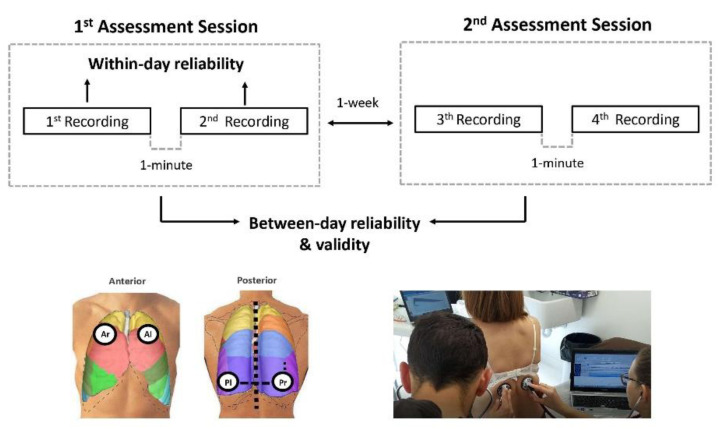
Study design. Overview of respiratory sounds collection for within-day and between-day reliability and construct validity. Anterior (**right** and **left**) and posterior (**right** and **left**) chest locations were recorded simultaneously for 25 s using two electronic stethoscopes. Within-day reliability was analysed using the two recordings from the first assessment session. Between-day reliability was analysed using the mean values of assessment session 1 (1st and 2nd recording) and the mean values of assessment session 2 (3rd and 4th recording). Construct validity was analysed using the mean values of the four recordings (1st, 2nd, 3rd and 4th recording) performed in both assessment sessions. Ar, anterior (**right**); Al, anterior (**left**); Pl, posterior (**left**); Pr, posterior (**right**).

**Figure 2 jcm-11-07509-f002:**
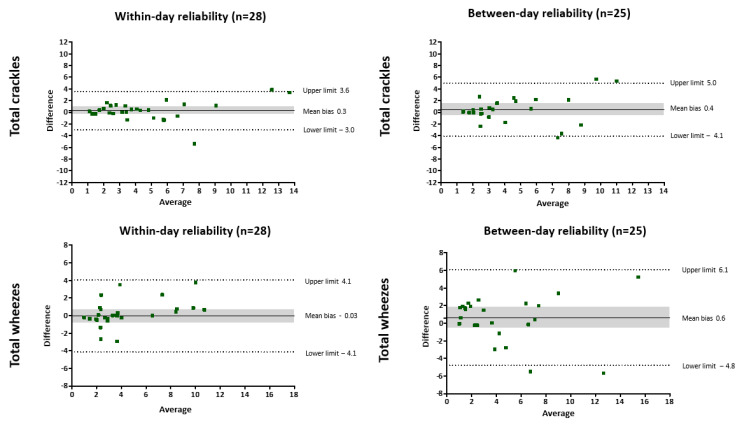
Bland-Altman plots for total number of crackles, wheezes and occupation rate considering both respiratory phases for within-day reliability (1 min apart) and between-day reliability (1 week apart). The solid line represents the mean difference between both measurements, dotted lines represent the 95% upper and lower limits of agreement and shared areas represent the 95% confidence interval for mean difference.

**Table 1 jcm-11-07509-t001:** Participants’ characteristics.

	Within-Day Analysis (*n* = 28)	Between-Day and Validity Analysis (*n* = 25)
Age, years, mean (SD)	62 (12)	61 (12)
Male, *n* (%)	9 (32)	7 (28)
BMI (kg/m^2^), median [P_25_–P_75_]	24.9 [22.0–26.4]	24.9 [21.2–26.4]
Aetiology of bronchiectasis, *n* (%)		
Post-infection	11 (39)	9 (36)
Idiopathic	7 (25)	6 (24)
Associated other respiratory disease	6 (21)	6 (24)
Others	4 (15)	4 (16)
FEV_1_ (% predicted), mean (SD)	77 (26)	78 (24)
Exacerbations in the last year, median [P_25_–P_75_]	4 [2,3,4,5]	4 [2,3,4,5]
Radiological extension *, *n* (%)		
≥3 lobes affected	20 (71)	17 (68)
BSI score (0–26), median [P_25_–P_75_]	7.5 [5.0–12.0]	7.0 [5.0–12.0]
BSI Classification, *n* (%)		
Mild (0–4)	5 (18)	5 (20)
Moderate (5–8)	13 (46)	12 (48)
Severe (≥9)	10 (36)	8 (32)
Long-term inhaled steroid treatment, *n* (%)	14 (50)	12 (48)
Long-term nebulised muco-active treatment, *n* (%)	4 (14)	3 (12)
Long-term antibiotic treatment, *n* (%)		
Oral	11 (39)	10 (40)
Nebulised	9 (32)	8 (32)
24 h sputum volume (mL), median [P_25_–P_75_]	11.0 [4.7–25.0]	11.2 [4.8–25.7]
QoL-B-Respiratory symptoms (0–100), mean (SD)	64.8 (17.7)	64.7 (18.4)

BMI, body mass index; P_25_–P_75_, percentile25–percentile75; *n*, number; SD, standard deviation; FEV_1_, forced expiratory volume in one second; BSI, bronchiectasis severity index; QoL-B, quality of life of bronchiectasis. * The last high-resolution computed tomography scan was performed within the previous 4 years before starting the study in all participants. Both cohorts were balanced.

**Table 2 jcm-11-07509-t002:** Within-day and between-day reliability of crackles during the inspiration, expiration and complete respiratory cycle and per chest location.

**Within-Day Crackles Reliability (*n* = 28)**									
	All Chest Locations			Anterior Chest Locations			Posterior Chest Locations		
	ICC (95% CI)	SEM	SDC	ICC (95% CI)	SEM	SDC	ICC (95% CI)	SEM	SDC
*Total number of crackles*									
Inspiratory phase	0.66 (0.39–0.83)	0.58	2.12	0.79 (0.60–0.90)	0.60	2.14	0.42 (0.05–0.68)	0.99	2.76
Expiratory phase	0.79 (0.60–0.90)	1.24	3.08	0.55 (0.24–0.76)	2.48	4.37	0.89 (0.79–0.95)	0.80	2.47
Complete respiratory cycle	0.87 (0.74–0.94)	1.26	3.11	0.74 (0.51–0.87)	2.19	4.10	0.77 (0.55–0.88)	1.58	3.49
**Between-day crackles reliability (*n* = 25)**									
	All chest locations			Anterior chest locations			Posterior chest locations		
	ICC (95% CI)	SEM	SDC	ICC (95% CI)	SEM	SDC	ICC (95% CI)	SEM	SDC
*Total number of crackles*									
Inspiratory phase	0.85 (0.69–0.92)	0.46	1.89	0.73 (0.49–0.87)	0.68	2.28	0.76 (0.53–0.88)	0.73	2.38
Expiratory phase	0.55 (0.22–0.77)	1.48	3.37	0.43 (0.05–0.70)	2.04	3.96	0.67 (0.39–0.84)	1.44	3.32
Complete respiratory cycle	0.70 (0.43–0.86)	1.81	3.73	0.64 (0.35–0.82)	2.10	4.02	0.74 (0.50–0.88)	1.84	3.76

ICC, Intraclass correlation coefficient; CI, confidence interval; SEM, standard error of measurement; SDC, smallest detectable change. Interpretation of ICC: excellent > 0.75, moderate between 0.4–0.75 and poor < 0.4.

**Table 3 jcm-11-07509-t003:** Convergent validity. The relationship between the mean values of ARS (all chest locations) recorded in both assessment sessions and clinical measures (*n* = 25).

	FEV_1_% Pred.	Sputum (mL)	LCQ	QoL-B Resp.	Radiological Severity ^#^	BSI
*Total number of crackles*						
Inspiratory phase	0.13	0.41 *	−0.14	0.27	0.04	0.27
Expiratory phase	0.08	0.26	0.09	0.35	−0.04	−0.31
Complete respiratory cycle	0.11	0.41 *	−0.0	0.31	−0.01	−0.33
*Total number of wheezes*						
Inspiratory phase	−0.05	0.53 *	0.00	0.04	−0.02	−0.36
Expiratory phase	−0.00	0.46 *	−0.02	0.12	−0.11	−0.34
Complete respiratory cycle	0.00	0.46 *	−0.0	0.11	−0.10	−0.35
*Occupation rate (%)*						
Inspiratory phase	0.04	0.48 *	−0.02	−0.02	−0.05	−0.32
Expiratory phase	−0.02	0.42 *	−0.05	0.03	−0.10	−0.27
Complete respiratory cycle	−0.01	0.44 *	−0.04	0.03	−0.03	−0.35

FEV_1_% pred., predicted value of forced expiratory volume in the first second; LCQ, Leicester cough questionnaire; QoL-B Resp., quality of life of bronchiectasis—Respiratory domain; BSI. bronchiectasis severity index. **^#^** The last high-resolution computed tomography scan was performed within the previous 4 years before starting the study in all participants. Interpretation of convergent validity: weak (r ≤ 0.29), moderate (0.30 < r > 0.59), strong (r ≥ 0.60); * *p* < 0.05.

**Table 4 jcm-11-07509-t004:** Within-day and between-day reliability of wheezes during the inspiration, expiration and complete respiratory cycle and per chest location.

**Within-Day Wheezes Reliability (*n* = 28)**									
	All Chest Locations			Anterior Chest Locations			Posterior Chest Locations		
	ICC (95% CI)	SEM	SDC	ICC (95% CI)	SEM	SDC	ICC (95% CI)	SEM	SDC
*Total number of wheezes*									
Inspiratory phase	0.81 (0.64–0.91)	0.61	2.17	0.79 (0.56–0.88)	0.82	2.52	0.74 (0.51–0.87)	0.56	2.08
Expiratory phase	0.86 (0.71–0.93)	0.90	2.63	0.85 (0.71–0.93)	0.89	2.62	0.73 (0.49–0.86)	1.51	3.40
Complete respiratory cycle	0.87 (0.73–0.94)	1.33	3.20	0.83 (0.67–0.92)	1.61	3.51	0.81 (0.63–0.91)	1.66	3.57
*Occupation rate (%)*									
Inspiratory phase	0.79 (0.60–0.90)	6.97	7.32	0.85 (0.69–0.93)	7.20	7.44	0.54 (0.22–0.76)	9.97	8.75
Expiratory phase	0.84 (0.69–0.92)	8.88	8.26	0.77 (0.54–0.88)	11.85	9.54	0.72 (0.48–0.86)	13.28	10.10
Complete respiratory cycle	0.86 (0.71–0.93)	6.96	7.31	0.83 (0.63–0.92)	8.58	8.12	0.73 (0.50–0.87)	10.65	9.05
**Between-day wheezes reliability (*n* = 25)**									
	All chest locations			Anterior chest locations			Posterior chest locations		
	ICC (95% CI)	SEM	SDC	ICC (95% CI)	SEM	SDC	ICC (95% CI)	SEM	SDC
*Total number of wheezes*									
Inspiratory phase	0.59 (0.26–0.80)	1.02	2.81	0.45 (0.09–0.71)	1.63	3.54	0.48 (0.11–0.74)	0.79	2.47
Expiratory phase	0.77 (0.54–0.89)	1.15	2.97	0.81 (0.61–0.91)	1.22	3.06	0.52 (0.16–0.75)	1.73	3.65
Complete respiratory cycle	0.78 (0.56–0.90)	1.83	3.75	0.74 (0.50–0.88)	2.50	4.38	0.57 (0.23–0.78)	2.30	4.20
*Occupation rate (%)*									
Inspiratory phase	0.58 (0.21–0.80)	10.56	9.01	0.54 (0.20–0.76)	14.51	10.56	0.41 (0.05–0.69)	10.52	8.99
Expiratory phase	0.74 (0.39–0.89)	11.52	9.41	0.60 (0.29–0.80)	15.94	11.07	0.71 (0.41–0.86)	12.92	9.96
Complete respiratory cycle	0.71 (0.33–0.87)	10.61	9.03	0.59 (0.27–0.80)	14.73	10.64	0.66 (0.34–0.84)	11.43	9.37

ICC, Intraclass correlation coefficient; CI, confidence interval; SEM, Standard error of measure; SDC, smallest detectable change. Interpretation of ICC: excellent > 0.75, moderate between 0.4–0.75 and poor < 0.4.

## Data Availability

The data that support the findings of this study are available from the first author (B.H-C), upon reasonable request.

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
