# Peer review of "Reliability and Validity of Computerized Adventitious Respiratory Sounds in People with Bronchiectasis"

_jcm, 2022, doi:10.3390/jcm11247509_

Round 1

Reviewer 1 Report

This study assessed the interest of using ARS in individuals with stable bronchiectasis. The authors assessed the reliability and validity (by correlation with other relevant outcomes) of ARS. They found, on average, moderate within- and between-day reliability. Moderate correlation was observed between ARS and sputum volume. 

I command these authors for this very well described and interesting study. The findings merit to be disseminated to the scientific-clinical community pending some changes that I believe can increase the quality of the paper.

Abstract

"[...] however, wheeze occupation rate only showed moderate reliability in the within-day analysis (ICC 27 0.86,95%CI 0.71-0.93)." Please rephrase, as this sentence do not explain how is the between-day reliability of this outcome. Suggestion: wheeze occupation rate showed moderate within-day reliability (ICC 0.86,95%CI 0.71-0.93) but poor between-day reliability (ICC ...)

Reading the abstract, one may feel that only the correlation between sputum volume and ARS has been tested with positive results. However, the correlation with other important outcomes has also been tested. Consider adding that there was no correlation between ARS and other PROM or severity indices.

The conclusion about the use of ARS in assessing PR components is weird. First, this perspective is not adequate givent the background provided in the abstract. Second, sequential ARS measurements could be also interesting in situations other than PR, such as during ACT which could be used separately from PR.

Introduction

We described, no comment, I enjoyed reading this part.

Methods

Please define if you recruited only non-CF bronchiectasis, especially since you state the need to distinguish CF from non-CF bronchiectasis in the discussion. If you recruited CF, this should be clearly explained and described in table  1.

How sputum volume has been collected 24-h before the first session since, as explained in the methods, participants were seen for the first time at the first session. Please clarify.

The threshold used to define the level of reliability is not consistent with the abstract, results, and discussion. For instance, occupation rates of wheezes is always above 0.4, so should be considered as moderate. Please also remain consistent in your terms (eg. use low or poor but not a mix).

Why Spearman's rank correlation coefficient has been chosen by default. Would Pearson correlation coefficient not be used instead? It may increase the power of correlations.

Results

A better understanding of the population is needed in table 1 as this is a very heterogeneous disease. For instance, it would have been very interesting to distinguish people with localised Bx from individuals with diffuse Bx (in >=2 lobes or everywhere in both lungs). Sensitive analysis according to these subgroups could yield to different conclusions. In addition, because all patients have a CT-scan, a correlation of ARS with the Bx extension should be provided as this could be an additional interesting outcome.

Within day reliability of crackles: What does mean "acceptable limits of agreements (LA)"? In abstract, "wide LA" is used. 

Same comment for between day reliability of crackles, and reliability of wheeze. "wide", "acceptable", etc is not defined in the methods, so it is only subjective. Please remain consistent in your terms and define, if possible, what do you consider as acceptable, wide, and so on.

In the same vein, "Bland-Altman plots showed good agreement for number of wheezes and occupation rate (figure 2). What does mean good agreement ? No bias ? Narrow LA ? please clarify.

Plots: please add the x-axis title for each pot. In addition, it is interesting to observe that below 7 crackles, the reliability is rather very good. Could it be interesting to comment on this? 

Discussion

Well described, I enjoyed reading it. Just one comment for the limitation section. The last limitation is a very interesting one. Can the authors estimate the human time that was used per recordings ? Most importantlly, had this human review of automatic recordings made any difference in the results ? This should be described in the methods and results.

Author Response

Please, find attached the file with our comments.

Reviewer 2 Report

More information should be given on the clinical manifestations of bronchiectasis and provide information on the type of burden they cause on the health system, such as hospitalizations or antibiotic use.

The time window prior to the study in which the primary diagnosis of bronchiectasis using a high-resolution computed tomography scan was performed is not specified.

It might have been interesting to compare the results with age and sex matched controls, without known lung pathology.

One of the objectives of the study that should have been reflected in the results is the definition of normality or baseline status of bronchiectasis lung in terms of ARS (crackles and wheezes) .

The sample size is reduced. Including smokers without MPOC criteria in the study to subsequently perform a separate analysis of the results obtained in the ARS (crackles and wheezes), would have been interesting.

Author Response

(The authors gave the same response as above.)

Round 2

Reviewer 1 Report

Dear authors,

Thank you for having considered all my comments. This new version looks perfect to me. Good job.

I have now understood that you used the the lower bound of 95%CI of ICC to define the reliability. This is very conservative. The readers will appreciate this choice, but I think it should be emphasided somewhere by adding a few words in the methods and/or the discussion, although I know you are limited by the word count. I think this is important because if we read the paper quickly, this information will likely be missed. However, it is possible that by increasing the number of participants, the 95% CI will become narrower, and many of your outcomes may change from poor to moderate. This is only a minor comment/suggestion.

Reviewer 2 Report

Thank you for the effort made to modify the article. 

Our point of view is to reconsider the publication of your work in the actual   evolutionary status of the research. The objective about the usefulness of the instrument to monitor post-physiotherapy of the patient with bronchiectasis ought to be sustained by a longitudinal study.